# COVID-19 Social Restrictions’ Impact on the Health-Related Physical Fitness of the Police Cadets

**DOI:** 10.3390/healthcare11131949

**Published:** 2023-07-05

**Authors:** Eduarda Sousa-Sá, Sara Pereira, Pedro B. Júdice, Luís Monteiro, Luís Miguel Massuça

**Affiliations:** 1Research Centre for Sport, Physical Education, Exercise and Health, Lusófona University, 1749-024 Lisbon, Portugal; 2Physical Activity, Health and Leisure Research Centre, Faculty of Sport, University of Porto, 4200-450 Porto, Portugal; 3ITR, Laboratory for Integrative and Translational Research in Population Health, 4050-600 Porto, Portugal; 4Centre for Research, Training, Innovation and Intervention in Sport, Faculty of Sport, University of Porto, 4200-450 Porto, Portugal; 5ICPOL, Research Centre, Higher Institute of Police Sciences and Internal Security, 1300-352 Lisbon, Portugal

**Keywords:** fitness, physical activity, police, COVID-19, lockdown

## Abstract

We aim to examine the changes in health-related physical fitness components, before and after COVID-19 social restrictions, in Police Academy cadets by age, sex, and physical activity (PA) level. A longitudinal analysis of 156 cadets (29.5% women) aged 25.4 ± 5.3 years old was conducted. All variables were collected before and after the lockdown period (13 weeks). PA levels were assessed with a self-reported questionnaire. Health-related physical fitness components were assessed based on the standardized protocols of morphological evaluation, speed, agility, strength, flexibility, and aerobic capacity tests. Despite the high correlations between pre- and post-pandemic social restrictions, we found significantly higher values for anthropometric indicators and lower physical fitness levels in post-pandemic restrictions, except for lower-body strength. When stratifying the sample by sex, age, and PA categories, the results indicate that the COVID-19 lockdown tends to differently impact anthropometric indicators and the physical fitness of the cadets, according to their sex, age, and PA categories. Our findings show that our sample reduces several health-related physical fitness components due to the social lockdown, with emphasis on cardiorespiratory fitness in men and upper-limb strength in women, highlighting the need to create effective strategies to keep police officers active during situations of less physical work.

## 1. Introduction

The evidence has shown that physical activity and physical fitness are health-promoting behaviors [1]. Physical activity is defined as any bodily movement produced by skeletal muscles that results in energy expenditure, whereas physical fitness is defined as the ability to conduct daily tasks with vigor and alertness, without undue fatigue and with ample energy to enjoy leisure-time pursuits and to meet unforeseen emergencies. The health-related components of physical fitness are cardiorespiratory fitness, muscular endurance and strength, body composition, and flexibility [2].

A recent review of systematic reviews showed a dose–response relationship between physical activity and premature mortality, as well as with the primary and secondary prevention outcomes of several chronic medical conditions (such as, type-2 diabetes, cardiovascular disease, hypertension, and all-cancer mortality, among others). This review also highlighted that, in unfit individuals, a slight increase in physical fitness levels led to a significant upturn in health status, with a decrease in the risk for chronic disease and premature mortality [3]. Due to the differences in body size and muscle mass, men and women usually portray different achievements when it comes to physical fitness performance. The evidence has shown that men have higher strength (muscular and endurance) and power (aerobic and anaerobic) levels [4,5].

Some occupations are physically more demanding than others and, therefore, need fitter workers. Police officers are one of those workers that need to invariably have appropriate fitness levels to deal with constantly physically demanding situations, as evidence has shown that poor motor fitness limits performance, threatening safety [6]. Therefore, a thorough insight into police officers’ levels of physical fitness can help to detect their strengths and weaknesses, realigning some practices, together with targeted interventions in this professional group.

In 2020, the spread of COVID-19 resulted in a pandemic. Evidence has shown that COVID-19 can cause a form of severe acute respiratory syndrome, which can rapidly lead to the death of vulnerable people, with mortality rates ranging from 6% to 15% [7]. Similar to other countries, Portugal also implemented social-distancing measures to limit the COVID-19 person-to-person contamination, generally including staying at home, except for essential medical care, food, or medicine shopping purposes.

The public health recommendations to prevent COVID-19 from spreading, specifically the closures of parks, gymnasiums, and fitness centers, together with stay-at-home orders, have the potential to sharply decrease daily physical activity, which is already documented in several countries [8,9]. The solution to stop the virus poses a great risk of reduced physical activity, with potential long-term health consequences since physical activity’s impact on non-communicable diseases is well-documented [1], as is the boost to the immune system, influencing the bodies’ anti-viral defenses [10].

People with better fitness levels also present better cardiovascular disease exercise-stress testing outcomes [11]. Recent research indicated that physical inactivity might be associated with increased COVID-19 severity [12], while individuals with moderate and high fitness levels showed a significantly lower risk of death from COVID-19 than those who were unfit [13]. Physical activity might be of help in fighting COVID-19 infection, as daily exercise enhances our immune system, thwarting some of the co-morbidities, such as diabetes, obesity, hypertension, and serious heart conditions, which make us more vulnerable to severe COVID-19 illness [10].

To the authors best knowledge, there are no studies, at present, observing the differences in health-related physical fitness components, before and after COVID-19 social restrictions, in the police force. Given the substantial impact that physical fitness shows on the quality of police activities, the aim of this investigation is to examine pre- to post-lockdown changes in health-related physical fitness components, due to COVID-19 social restrictions, in Police Academy cadet students by age, sex, and physical activity level.

## 2. Materials and Methods

This was a longitudinal study with a convenient sample of Police Academy cadet students from the Higher Institute of Police Sciences and Internal Security in a situation of home confinement (3 months). The sample consisted of 156 cadet students from the Higher Institute of Police Sciences and Internal Security (1st, 2nd, 3rd, and 4th years of the Integrated Master in Police Science).

### 2.1. Physical Activity Perception in a Context of Social Restrictions

Physical activity levels were assessed using a self-reported questionnaire (by asking the participants to recall their physical activity habits during the COVID-19 lockdown—May 2020), and 3 categories were considered (less active, moderately active, and active).

### 2.2. Physical Fitness Evaluations

The participants were assessed in two time points: moment 1—before COVID-19 (late January 2020), hereafter called ‘pre-restrictions’; and moment 2—after the in-person return to ISCPSI (late June 2020), hereafter called ‘post-restrictions’. When returning to the in-person lessons, it was imperative to strictly follow the guidelines of the Directorate-General of Health, and the order of the tests’ performance was the same as for ‘pre-restrictions’.

*Anthropometric indicators.* Weight (kg) and height (cm) were measured according to protocol [14]. Weight was measured to the nearest 0.5 kg, using a Secca body scale (Vogel & Halke, Hamburg, Germany), and height was measured to the nearest mm (0.1 cm) using a Siber-Hegner anthropometric kit (DKSH Ltd., Zurich, Switzerland). Body mass index was computed using the standard formula. Individual measurements were collected for all participants by the same evaluators (intra-observer technical error of measurements: height, R ≥ 0.98).

*Fitness evaluations.* All participants completed eight fitness tests, from which seven variables were collected for the analysis. Speed was assessed with a 30 m run test. The participants completed three trials and the best score (time in seconds) was recorded for the analysis [15]. All sprint times were recorded using electronic timing lights (Wireless Sprint System, Brower Timing Systems, Draper, UT, USA). Agility was assessed with the slalom test, which required the subject to run a course in the shortest possible time. A standard slalom course was composed of four cones (A, B, C, and D) placed according to Figure 1, and the test (i) began at A, then a straight line to D, and then back to A; (ii) a standard zig-zag course from A to B, C, D, then back to C, B, and A; (iii) then a straight line from A to D, then back to A; and (iv) then a straight line from A to E (finish). The test stopped when the participant: (i) did not start from the stopped position; (ii) did not perform the test course in the correct order; (iii) did not circle the cone (on the outside); or (iv) touched or knocked down any cone. The participants completed two trials and the time (in seconds) was recorded using electronic timing lights (Wireless Sprint System, Brower Timing Systems, USA). The best score was recorded for the analysis. The test stopped when the subject did not follow the protocol’s rules. Please see Appendix A (Appendix A) for a further explanation of the test. Lower-body strength was assessed by the horizontal impulsion test (standing broad jump). The subject was asked to stand behind the starting line with feet parallel and to jump as far as possible. The best of two trials was recorded and the maximum distance was measured to the nearest cm [16]. Abdominal strength (and endurance) was measured by the 60 s sit-up test [17]. At the bottom position, the shoulder blades had to touch the ground and, at the top, the elbows had to touch the knees. The start and finish commands were given by the investigator, who registered the number of repetitions. Participants were allowed to rest in the ‘down’ position; however, only complete repetitions were counted. The participants completed one trial and the number of repetitions was recorded. Static strength was measured through a maximal isometric handgrip-strength test using a dynamometer (Takei Physical Fitness Test, TKK 5001, GRIP–A, Tokyo, Japan). The participants performed the test twice with each hand and the sum of the best results achieved by both hands was considered (in kg ^f^) [18]. Flexibility was assessed by the sit and reach test [19]. Each subject was seated barefoot on the floor with their legs stretched out straight ahead and with their feet placed with the soles flat against the sit and reach box. The test was performed as per protocol. The score of the test was recorded to the nearest centimeter as the distance reached by the fingertips. Aerobic capacity was assessed with the shuttle test [20]. The participants ran back and forth between two lines, 20 m apart at 8.5 km/h, with the speed increasing by 0.5 km/h/min. The test continued until the participants reached exhaustion or could not complete the laps twice, continuously, within the required time limit. The total laps at the final stage and the estimated maximal oxygen uptake (VO_2_ max, by applying the equation proposed by Ramsbottom, Brewer [21]) were considered as indicators of performance.

### 2.3. Statistical Analysis

Descriptive statistics (mean and standard deviation) were used to analyze the participants’ performances for the anthropometric indicators and physical fitness tests. To test the mean changes between pre- and post-pandemic social restrictions, we used the parametric Student’s *t*-test for paired samples. To test the stability of the variables in the two time points, we used Pearson’s correlation. All analyses were performed by sex and age (divided into three groups: 18–24; 25–30; and 31–36 years old). Individual test results were transformed into *z*-scores allowing for the comparison of dissimilar metrics and to plot a profile for pre- and post-pandemic social restrictions. Care was taken to reverse signs in weight and body mass index; hence, in both indicators, lower values indicated better health. The same logic applied for speed and agility tests, in which less time represented a better performance. Finally, the influence of the participants’ self-reported physical activities during the lockdown period on physical fitness tests during post-pandemic social restrictions were compared using ANCOVA, with age as a covariate. Moreover, the Bonferroni test was used for post hoc multiple comparisons. All the data analyses were performed using the Statistical Package for the Social Sciences (SPSS Statistics version 27.0, IBM Corp., Armonk, NY, USA), and the significance level was set at 5%.

## 3. Results

Our sample comprised 156 participants (29.5% women) aged 25.4 ± 5.3 years old. The changes in anthropometric and physical fitness indicators between pre- and post-pandemic restrictions are presented in Table 1. All variables presented a moderate to high correlation from pre- to post-pandemic restrictions (*p* < 0.05); however, we observed significantly higher values for anthropometric indicators and lower physical fitness levels during post-pandemic restrictions compared to moment 1 (pre-pandemic restrictions), except for lower-body strength, where the participants did not change their mean values (*p* > 0.05).

The differences between pre- and post-pandemic restrictions by sex are provided in Figure 1 (and Appendix A). In the entire sample, a moderate to high correlation in all variables for both women and men was observed; although, women displayed higher values for anthropometric indicators and speed, and lower values for body strength than men. Conversely, men had higher correlation values in the static and abdominal strength tests. Moreover, women did not show differences in anthropometric indicators, flexibility, and abdominal strength between pre- and post-restrictions and men in speed and lower-body strength (*p* > 0.05).

We also analyzed the differences in anthropometric indicators and physical fitness during pre- and post-pandemic restrictions by age, which can be observed in Figure 2 (and Appendix A). Regarding the anthropometric indicators, the stability results were higher for all ages (ρ from 0.896 to 0.957). However, when testing the mean differences between pre- and post-pandemic restrictions, we only found differences in the oldest group, with higher values for weight and body mass index during the post-pandemic restrictions. Our physical fitness results indicate that abdominal strength is the more unstable component in all age groups (ρ from 0.537 to 0.577). However, a different trend in the mean change values during pre- to post-pandemic restrictions was found depending on age, except for lower-body strength, which did not change in all age groups. More specifically, the youngest students did not change their mean values for speed; students aged 25–30 years old maintained their physical fitness levels for agility and abdominal strength, while the oldest students kept their mean values for abdominal strength.

Finally, we sought to understand if the participants with different self-reported physical activities during the lockdown differed in their anthropometric indicators and physical fitness levels during post-pandemic restrictions (Table 2). For weight and body mass index, the ‘less-active’ group had higher values (76.3 ± 1.8 kg and 24.8 ± 0.4 kg/m^2^, for weight and body mass index, respectively) than the ‘moderately active’ group (70.0 ± 1.3 kg and 22.9 ± 0.3 kg/m^2^, for weight and body mass index, respectively). Concerning physical fitness levels, we only found differences in cardiorespiratory fitness and VO_2_ max, favoring the ‘active’ group compared to the ‘less-active’ and ‘moderately active’ groups. No differences between the ‘moderately active’ and ‘active’ groups were found.

## 4. Discussion

Governments have been struggling to control and limit the virus’ spread, with most countries adopting social confinement (partial or generalized lockdowns) as the main strategy to fight this pandemic. The evidence shows that the lockdown has driven society to move less, with a subsequent decrease in physical activity levels [22], which ultimately affects the levels of physical fitness [23,24]; however, no investigation has considered social restrictions’ impact on the police forces’ physical fitness yet, a group for which physical fitness is paramount.

Our findings from a sample of Police Academy cadet students showed a significant deterioration in several health-related physical fitness components, developed during the lockdown, with a special emphasis on cardiorespiratory fitness, in which we found an overall decrease of about 3 mL/kg/min in just 13 weeks. This aligns with the findings from a previous investigation where a decrease in cardiorespiratory fitness was also found (5.6 mL/kg/min) after two months of confinement. The evidence seems to suggest that, even when exercise training was performed during the lockdown, athletes (perhaps the most similar sample to ours) decreased their cardiorespiratory fitness as well [25,26]. Both these groups are highly dependent on their fitness levels, as a low aerobic capacity limits their professional performance.

In addition to cardiorespiratory fitness, we also found that the social lockdown had a negative impact on other fitness components, such as body mass index, strength, and speed. The findings for professional soccer athletes also show decreases in these fitness components after a period of social lockdown [27], with one investigation showing a 5% decrease in maximal running speed after two months of confinement [25] and another study showing an increase in 10 and 20 m sprint times, as well as a decrease in countermovement high jump, after 63 days of confinement. Similarly, in our study, the cadets from the police force also lost speed and agility due to the confinement and, although small changes occurred in lower-limb strength, we found a significant decrease in the horizontal jump. Moreover, our study adds knowledge to the previous findings by showing that all types of strength are negatively affected by the lockdown (isometric, abdominal, lower-limb strength). A recent investigation found that three weeks of detraining did not affect muscle thickness, strength, or sport performance [28]. However, this study was performed on adolescent athletes, which, together with the short period of a lack of exercise, could potentially explain the contradicting findings, as older individuals are more prone to lose their physical fitness levels. Another recent investigation on young adult endurance runners found that two weeks of detraining was enough to reduce cardiopulmonary functions and muscle strength [29], which somewhat aligns with our findings.

Interestingly, our results show that women suffer a more pronounced decrease in speed, agility, and lower-limb strength compared to men, whereas cardiorespiratory fitness is the most affected fitness component for men. There is a lack of research on the distinct sex-related responses related to inactive periods, with the available evidence showing that both men and women reduced their cardiorespiratory fitness levels in these situations [30]. Moreover, there are some indications that men tended to lose more cardiorespiratory fitness from aging than women [31], which somehow reinforced our findings. Concerning the loss of strength from disuse in both men and women, the existing evidence is not clear regarding the role of sex; however, it seems that women may be more susceptible to lose strength than men [32], when subjected to inactive periods. Considering the distinct responses to atrophic stimuli between sexes, also shown in our results, further research investigating the different mechanisms for muscular atrophy among sexes is warranted.

Regarding age, our findings suggest that older individuals present greater losses of fitness, when compared to the younger ones, as it can be observed by the effect sizes. The decrease in VO_2 max_ in the older group had an effect size of 0.80, whereas in the younger group, the decrease in VO_2 max_ had an effect size of 0.64. Likewise, for upper-body strength, the effect size for the reduction in the older group was 0.96, whereas in the younger group, it was only 0.49. Furthermore, even though all age groups increased their body mass index from pre- to post-lockdown, the older the age group, the greater the increase in the body mass index. This meant that, along with decrements in cardiorespiratory fitness and muscle strength, the older individuals’ body mass indexes were negatively impacted by the lockdown, which had implications regarding performance, but most importantly health. This reinforces the previous findings showing that, with age, both cardiorespiratory and strength fitness levels tend to decrease at a higher magnitude, due to inactive periods [31]. Thus, in times of lockdown, more attention must be granted to the older age groups, as they will be more impacted by these periods, which may exacerbate the natural losses that already occurred and compromise their health.

Finally, our findings show that participants perceiving their physical activity levels as low during the lockdown were the ones presenting poorer fitness levels in the post- lockdown period, while the active ones during the isolation period presented significantly higher fitness levels. These results were also found for cardiorespiratory fitness and upper-limb strength, with no differences in the other fitness components between groups with different perceived physical activity levels. This confirms that, even in highly trained men and women, performing some physical activity during the lockdown may prevent losses in fitness components, which highlight the need for strategies to maintain physical activity levels during these pandemic situations in the future.

To our knowledge, this is the first investigation that observed the differences in health-related physical fitness components, before and after COVID-19 social restrictions in the police force. Moreover, we objectively assessed different components of health-related fitness (which is uncommon). Additionally, for a longitudinal study of this nature, our sample size was much larger than the previous investigations mostly observing small groups of athletes. Nevertheless, this investigation is not without its limitations. The self-reported assessment of physical activity during the social lockdown entailed subjectivity and may have generated some bias in the analyses. However, the fact that self-reported physical activity was positively related with the objectively measured cardiorespiratory fitness led us to conclude that, even acknowledging some limitations, the physical activity assessment was somehow valid. Although we included both men and women, around 70% of the sample were men, thus limiting generalizations for the female police officers.

## 5. Conclusions

Our findings highlight that the Police Academy cadet students had several reduced health-related physical fitness components due to the lockdown, with emphasis on cardiorespiratory fitness in men and upper-limb strength in women. Our investigation highlighted that the lockdown itself was responsible for relevant and significant reductions in several fitness components, even when considering a sample of highly trained participants. It is urgent to create and disseminate effective strategies to maintain police officers’ physical activity levels during the pandemic to avoid important losses in fitness levels and concomitant health-related issues.

## Figures and Tables

**Figure 1 healthcare-11-01949-f001:**
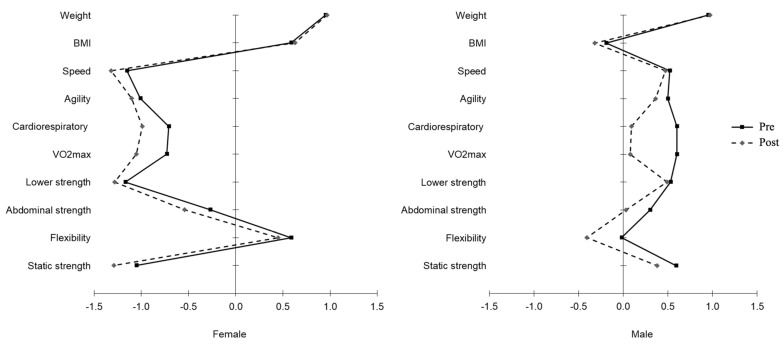
Graphical representation of the anthropometric indicators and physical fitness levels of pre- and post-pandemic restrictions by sex (all variables were standardized (*z*-scores)).

**Figure 2 healthcare-11-01949-f002:**
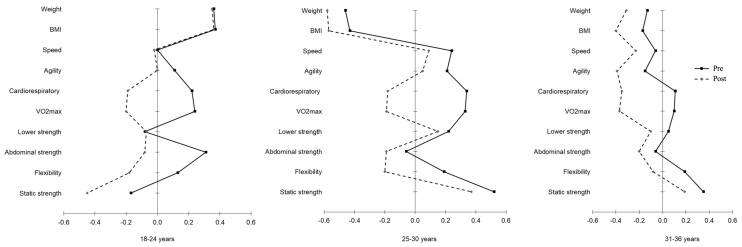
Graphical representation of the anthropometric indicators and physical fitness levels during pre- and post-pandemic restrictions by age (all variables were standardized (*z*-scores)).

**Table 1 healthcare-11-01949-t001:** Changes in anthropometric and physical fitness indicators between pre- and post-pandemic social restrictions for the whole sample.

Whole Sample (*n* = 156)					
	Pre	Post			
**Anthropometric indicators**	Mean ± SD	Mean ± SD	*r*	*t*	Effect size ^d^
Weight (kg)	71.3 ± 10.4	71.9 ± 11.2	**0.955**	**−2.570**	0.21
Body mass index (kg/m^2^)	23.3 ± 2.4	23.5 ± 2.7	**0.918**	**−2.498**	0.20
**Physical fitness**					
Speed (s)	4.4 ± 0.3	4.4 ± 0.4	**0.925**	**−2.678**	0.22
Agility (s)	23.1 ± 1.4	23.3 ± 1.5	**0.872**	**−3.610**	0.30
Cardiorespiratory fitness (no. laps)	80.4 ± 19.5	71.3 ± 19.1	**0.764**	**8.354**	0.68
VO_2 max_ (mL/kg/min)	45.9 ± 6.1	43.0 ± 6.1	**0.756**	**8.345**	0.69
Lower body strength (m)	2.3 ± 0.2	2.3 ± 0.3	**0.920**	**1.790**	0.15
Flexibility (cm)	53.4 ± 6.6	51.3 ± 7.4	**0.811**	**6.147**	0.50
Static strength (kg ^f^)	94.1 ± 21.7	89.1 ± 21.5	**0.920**	**6.613**	0.58
Abdominal strength (no. reps/60 s)	57.3 ± 6.8	55.7 ± 7.5	**0.547**	**2.910**	0.24

*r* = Pearson’s correlation; *t* = paired sample *t*-test; ^d^ = Cohen’s d; **bold** = *p* < 0.05. s = seconds; cm = centimeters; kg ^f^ = kilogram-force; no. = number; reps = repetitions.

**Table 2 healthcare-11-01949-t002:** Body composition and physical fitness levels according to physical activity categories during post-pandemic social restrictions.

	Less Active (*n* = 35)	Moderately Active (*n* = 65)	Active (*n* = 37)		
**Body composition**	Mean ± SE	Mean ± SD	Mean ± SD	Z	Post hoc comparisons
Weight (kg)	76.3 ± 1.8	70.0 ± 1.3	73.5 ± 1.7	**9.483**	LA > MA
**Body mass index (kg/m^2^)**	24.8 ± 0.4	22.9 ± 0.3	23.7 ± 0.4	**13.049**	LA > MA
Physical fitness					
Speed (s)	4.3 ± 0.7	4.4 ± 0.5	4.4 ± 0.6	0.472	
Agility (s)	23.4 ± 1.7	23.1 ± 1.4	23.1 ± 1.3	0.531	
Cardiorespiratory fitness (no. laps)	64.0 ± 16.4	70.3 ± 20.2	80.3 ± 18.0	**4.741**	LA < A; MA < A
VO_2 max_ (mL/kg/min)	40.7 ± 5.4	42.7 ± 6.4	45.8 ± 5.5	**4.771**	LA < A; MA < A
Lower body strength (m)	2.3 ± 0.3	2.3 ± 0.2	2.3 ± 0.2	0.115	
Flexibility (cm)	50.8 ± 6.3	50.4 ± 8.3	51.7 ± 6.2	0.252	
Static strength (kg ^f^)	97.2 ± 17.8	86.0 ± 20.9	91.4 ± 22.2	1.297	
Abdominal strength (no. reps/60 s)	54.2 ± 7.6	55.0 ± 7.5	57.7 ± 8.3	1.545	

**bold** = *p* < 0.05; LA = less active; MA = moderately active; A = active. s = seconds; cm = centimeters; kg ^f^ = kilogram-force; no. = number; reps = repetitions.

## Data Availability

The data presented in this study are available upon reasonable request from the corresponding author. The data are not publicly available due to privacy and ethical restrictions.

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
