# Peer review of "COVID-19 Social Restrictions’ Impact on the Health-Related Physical Fitness of the Police Cadets"

_healthcare, 2023, doi:10.3390/healthcare11131949_

Round 1
Reviewer 1 Report
The present study describes the effect of COVID-19 social restrictions impact on the health-related physical fitness of Police Cadets. As the authors comment, the study finding showed that several health-related physical fitness components were reduced due to the social lockdown, highlighting the need for effective strategies to keep police officers active during periods of time away from the workplace.
The study was based on a good and useful idea as it concerns the health and fitness of a group of workers, such as police officers, who need to be in excellent physical condition. However, it raises some issues about its design. Some factors are not clear, such as the duration of the police cadets' confinement at home and the time of testing.
2,3. Format the title text according to the journal's template and the journal's guidelines for authors.
2. Materials and Methods
77-78. (3 months). Was this the actual period of confinement at home in Portugal?
2.2. Physical fitness evaluations
85-89. Were the tests mandatory at the specific times they were taken?
There appears to be a considerable period during which the cadets were not inactive due to house confinement.
From the overall findings, it appears that the main factor that influenced the physical fitness of the police cadets was their usual activity level and age rather than confinement at home. Did you consider whether there was a correlation between physical activity and the age of the subjects?
97.... Fitness evaluations.
What was the recovery time between trials?
If you are referring to speed, it may be better to present it as m/s.
4. Discussion
198-214.
It would be useful to define terms more clearly, such as social confinement (partial or generalized lockdown), lockdown, social restrictions, confinement, home confinement, and social lockdown, according to the law, describing human activity differently.
It is probably better to clearly describe the allowable activity of individuals during the three months referred to.
Author Response
The present study describes the effect of COVID-19 social restrictions impact on the health-related physical fitness of Police Cadets. As the authors comment, the study finding showed that several health-related physical fitness components were reduced due to the social lockdown, highlighting the need for effective strategies to keep police officers active during periods of time away from the workplace.
The study was based on a good and useful idea as it concerns the health and fitness of a group of workers, such as police officers, who need to be in excellent physical condition. However, it raises some issues about its design. Some factors are not clear, such as the duration of the police cadets' confinement at home and the time of testing.
2,3. Format the title text according to the journal's template and the journal's guidelines for authors.
- Thank you for this suggestion. To the authors’ understanding, the title aligns with the journal’s guidelines.
- Materials and Methods
77-78. (3 months). Was this the actual period of confinement at home in Portugal?
- Thank you for this question. Yes, this was the confinement duration of the 1st lockdown in Portugal, which took place during the data collection of our study.
2.2. Physical fitness evaluations
85-89. Were the tests mandatory at the specific times they were taken?
- Thank you for this question. We are not sure if we understood this question. However, yes, all assessments were mandatory at the time of data collections.
There appears to be a considerable period during which the cadets were not inactive due to house confinement.
- Thank you for this comment. We are not sure to have understood this question. Nevertheless, our study only assessed the participants at two time-points, and physical inactivity was not included in the assessments. We subjectively assessed physical activity levels during the 3 months of the 1st As such, we have no way of reporting physical inactivity levels and their context.
From the overall findings, it appears that the main factor that influenced the physical fitness of the police cadets was their usual activity level and age rather than confinement at home. Did you consider whether there was a correlation between physical activity and the age of the subjects?
- Thank you for this question. To accommodate this reviewer’s suggestion, we have replaced the ANOVA for an ANCOVA to include age as a covariable, and the results remained similar. Please see statistical analysis subsection (lines 144-145) and the results section (lines 190-196 and table 2).
97.... Fitness evaluations.
What was the recovery time between trials?
- Thank you for this question. The recovery time was performed as per protocols of each test administrated – references are included in the manuscript.
If you are referring to speed, it may be better to present it as m/s.
- Thank you for pointing this out. The test is used to measure how many seconds the participant takes to run a specific and fixed number of meters – 30 metres. As such, the test does not evaluate the number of meters ran in a given number of seconds. The correct way to present the results, and following the test’s protocol, is seconds.
- Discussion
198-214.
It would be useful to define terms more clearly, such as social confinement (partial or generalized lockdown), lockdown, social restrictions, confinement, home confinement, and social lockdown, according to the law, describing human activity differently.
- Thank you for pointing this out. From lines 198-214, the terms mentioned by the reviewer are related to the entire world and not only to a single country (Portugal). Due to the lack of significant differences in their definitions across countries, the authors believe that adding such information to the manuscript would increase the difficulty in understanding the message being spoken and would not be directly related to the entire manuscript’s aims, as this work does not relate to social/governmental laws.
It is probably better to clearly describe the allowable activity of individuals during the three months referred to.
- Thank you for pointing this comment. This information is present in the introduction section, lines 51-53: “Like other countries, Portugal has also implemented social distancing measures to limit the Covid-19 person-to-person contamination, generally including staying at home, except for essential medical care, food or medicine shopping.”. The authors believe that this information is needed in the introduction section as it supports and explains the entire rationale and methodology of the study.
Reviewer 2 Report
* 233-238: I belive that you could make an inference about the gender difference for the kind of fitness decreased during the lockdown, based upon the results of previous studies investigating the difference in between how long males take to increase the level of a particular fitness and that for females.
Acceptable
Author Response
233-238: I belive that you could make an inference about the gender difference for the kind of fitness decreased during the lockdown, based upon the results of previous studies investigating the difference in between how long males take to increase the level of a particular fitness and that for females.
- Thank you for this suggestion. However, the authors believe to have somewhat addressed this point of view in the manuscript – lines 244-251 – “There is a lack of research on the distinct sex-related responses related to inactivity moments, with available evidence showing that both men and women reduce their cardiorespiratory fitness levels in these situations [28]. Still, there are some indications that men tend to lose more cardiorespiratory fitness from aging than women [29], which somehow reinforces our findings. Concerning the loss of strength from disuse in both men and women, the existing evidence is not clear regarding the role of sex, but it seems that women may be more susceptible to lose strength than men [30], when subjected to inactivity periods.”.
Nevertheless, we searched for more data to include, and in fact, there is not available evidence that can fully respond to this reviewer’s comment. The only study found, a longitudinal study (5-year) among state patrol officers, showed significant decreases among male officers in lower-body power, muscular endurance (push-up), and aerobic capacity. In contrast, significant improvements were observed in muscular endurance (sit-up) among female officers1. We believe that discussing these results with ours is not entire appropriate as this study’s duration is much longer than ours, therefore health-related changes might be completely different.
1 Dawes JJ, Lopes Dos Santos M, Kornhauser C, Holmes RJ, Alvar BA, Lockie RG, Orr RM. Longitudinal Changes in Health and Fitness Measures Among State Patrol Officers by Sex. J Strength Cond Res. 2023 Apr 1;37(4):881-886. doi: 10.1519/JSC.0000000000004327.
Reviewer 3 Report
The topic is relevant in the field, as we need to understand the effects of the lockdown on various aspects and populations. The study will provide valuable information on how to support and give guidance to police cadets to overcome the negative effect of the lockdown as well as to prepare for future pandemics.
With regards to the methodology, how was the physical activity perception measured? More specific information is needed for example how was the information given, per day, per hour? Was it for the entire lockdown or only for 1 day? Weekday or weekend day?
The other concern is how was it grouped in the 3 categories, on what basis? Was Mets values used? No information regarding that is explained in the section.
No mention in introduction regarding the differences between men and women and physical fitness variables.
No mention of any ethical approval or consent from participants to participate in this study?
Author Response
The topic is relevant in the field, as we need to understand the effects of the lockdown on various aspects and populations. The study will provide valuable information on how to support and give guidance to police cadets to overcome the negative effect of the lockdown as well as to prepare for future pandemics.
With regards to the methodology, how was the physical activity perception measured? More specific information is needed for example how was the information given, per day, per hour? Was it for the entire lockdown or only for 1 day? Weekday or weekend day?
- Thank you for these questions. Physical activity was self-reported via an online questionnaire, and the response options were active, less active and moderately active.
The other concern is how was it grouped in the 3 categories, on what basis? Was Mets values used? No information regarding that is explained in the section.
- Thank you for this question. As per the previous comment, physical activity perception was grouped in 3 categories according to the participants’ answers - active, less active and moderately active.
No mention in introduction regarding the differences between men and women and physical fitness variables.
- Thank you for this suggestion. We have added information on physical fitness differences between men and women, in the introduction section, lines 40-43: “Due to differences in body size and muscle mass, men and women usually portrait different achievements when it comes to physical fitness performance. Evidence has shown that men have higher strength (muscular and endurance) and power (aerobic and anaerobic) levels [4, 5]”.
No mention of any ethical approval or consent from participants to participate in this study?
- Thank you for this question. This information is present at the end of the manuscript, after the conclusions section, as per journal’s guidelines. Nevertheless, we added it here: “The study was conducted according to the guidelines of the Declaration of Helsinki and approved by the Ethics Committee of the Higher Institute of Police Sciences and Internal Security (Lisbon, Portugal)” and “Informed consent was obtained from all subjects involved in the study”.
Reviewer 4 Report
The work solves a specific research problem related to determining the impact of social restrictions related to lockdown during Covid on the physical fitness determining the level of health of cadets of the Police Academy. The results of the above studies conducted at the beginning of Covid and after its completion through a questionnaire and physical fitness tests, taking into account age, gender and level of physical activity, revealed significant differences in the examined characteristics. The study uses a sufficiently large research material, the correct methodology, as well as the appropriate statistical tools. The work has a correct layout, it is clear in its content in each chapter.
Author Response
The work solves a specific research problem related to determining the impact of social restrictions related to lockdown during Covid on the physical fitness determining the level of health of cadets of the Police Academy. The results of the above studies conducted at the beginning of Covid and after its completion through a questionnaire and physical fitness tests, taking into account age, gender and level of physical activity, revealed significant differences in the examined characteristics. The study uses a sufficiently large research material, the correct methodology, as well as the appropriate statistical tools. The work has a correct layout, it is clear in its content in each chapter.
- Thank you so much for this comment.